**Data Availability Statement:** All data are available. The link to all raw data is https://aact.ctti-clinicaltrials.org/connect.

# Changes in key recruitment performance metrics from 2008–2019 in industry-sponsored phase III clinical trials registered at ClinicalTrials.gov

**Mette Brøgger-Mikkelsen[1,2], John Robert Zibert[2], Anders Daniel Andersen[2], Ulrik Lassen[3], Merete Hædersdal[1], Zarqa Ali[1]\*, Simon Francis Thomsen[1,4]**

**1** Department of Dermato-Venereology, Bispebjerg Hospital, Copenhagen, Denmark, **2** Studies&Me A/S, Copenhagen, Denmark, **3** Department of Oncology, Rigshospitalet, Copenhagen, Denmark, **4** Department of Biomedical Sciences, University of Copenhagen, Copenhagen, Denmark

\* zarqa_ali@hotmail.com

## Abstract

### Background

Increasing costs and complexity in clinical trials requires recruitment of more narrowly defined patient populations. However, recruitment for clinical trials remains a considerable challenge.

### Aim

Our overall aim was to quantify recruitment performance in industry-sponsored phase III clinical trials conducted globally during 2008–2019 with primary aim to examine development of overall clinical trial measures (number of trials completed, number of participants enrolled, trial duration in months) and key recruitment metrics (recruitment rate, number of sites, number of patients enrolled per site).

### Methods

The publicly available AACT database containing data on all trials registered at Clinical-Trials.gov since 2008 was used. The analysis was completed during three time periods from 2008–2019 of 4 years each.

### Results and conclusion

Recruitment duration for industry-sponsored phase III clinical trials have increased significantly during the last 12 years from an average recruitment period of 13 months (IQR 7–23) in 2008–2011 to 18 months (IQR 11–28) in 2016–2019 (p = 0.0068). Further, phase III clinical trials have increased the number of registered sites per clinical trial by more than 30% during the last 12 years from a median number 43 sites (IQR 17–84) in 2012–2015 to 64 sites (IQR 30–118) in 2016–2019 (p = 0.025), and concurrently, the number of participants enrolled in clinical research has decreased significantly from 2012–2015 and 2016–2019

**Funding:** The authors received no specific funding for this work.

**Competing interests:** The authors have declared that no competing interests exist.

(p = 0.046). We believe that these findings indicate that recruitment for phase III clinical trials is less effective today compared to 12 years ago.

## Introduction

The complexity and costs of clinical trials have increased dramatically over the past decade, especially in the area of new drug development [1]. Increasing prevalence of chronic diseases, and increasing needs for personalised medicine and advanced technologies are just some of the demands that the clinical trial industry is currently facing [2]. New investigational drugs that target chronic, difficult-to-treat and rare diseases require recruitment of a more narrowly defined patient subpopulation, increasing the scope of clinical trials and the burden to execute them [3]. However, recruitment of participants into clinical trials remains a considerable challenge. In approximately 80% of clinical trials, enrollment timelines are not met [2, 4] and for each day of trial delays, pharmaceutical companies stand to lose from $600,000 to $8 mio. per day [5].

Recruiting patients for clinical research is a difficult task requiring sponsors to dedicate significant resources. The recruitment process not only includes identifying and screening potential participants by eligibility criteria before enrolling patients in the trial. During the process, potential participants should be thoroughly informed about all aspects of the trial by educated and relevant personnel, ensuring full comprehension and voluntariness and obtaining informed consent for participation [5]. Unfortunately, 6–29% of clinical trials terminate due to insufficient recruitment [6–8].

When recruitment is slow and inefficient it may have serious scientific, financial and ethical consequences for both patients and the clinical trial stakeholders involved [9]. Not only will study delays result in higher costs of the clinical trial, lengthy trials also delay the availability of potentially beneficial new treatments to the public. Moreover, participants who are occupied in ongoing trials are rarely eligible for other clinical studies, diminishing the pool of patients available for clinical trials. The integrity and validity of clinical trials is highly dependent on adequate sample sizes, as the risk of type II error (e.g., false negative discoveries) increases when target sample size is not achieved [10]. Ultimately, trials may be terminated due to low data quality wasting considerable human and material resources without a compensatory gain in research knowledge.

From an operational standpoint, the goal is to complete clinical trials as quickly as possible without delays and with the highest data quality [11]. However, such goals require efficient and high-performing clinical trial sites. Unfortunately, low- and non-performing sites that only enroll a single subject or no subjects at all, remain a significant challenge for recruitment and trial efficiency. According to multiple sources, the overall industry average of low or non-performing sites is approximately 33% on any given trial at a cost of approximately $50,000 per low or non-performing site [12, 13].

Previous poor recruitment performance, busy site staff, lack of research experience within site staff and slow approval processes are just some of the factors that may influence the performance of clinical trial sites [14]. In the literature, few studies have assessed clinical trial site performance. In a study evaluating 105 industry-sponsored phase II-IV clinical trials, Lamberti et al. found that 22% of the clinical trial sites did not enrol any participants [15]. Another study, investigating recruitment performance of clinical trial sites on a regional level, found that recruitment performance was lower than forecasted in Europe and North America,

whereas recruitment was 104% of predicted in Latin America despite ethics, regulatory and contractual delays [16]. These findings underline the complexity of site selection for clinical trials and the importance of rapid recruitment in order to meet enrollment timelines.

Phase III trials account for most of the resources spent during clinical testing with the top cost drivers of trial expenditures being clinical procedure costs (15–22% of total), administrative staff costs (11–29% of total), and site monitoring costs (9–14% total) [17, 18]. Phase III clinical trials are pivotal in proving long term safety and efficacy, however phase III trials are highly expensive with a median cost around $21 million for a single trial [19]. Consequently, successful recruitment in phase III clinical trials is particularly important. In this study, we investigated changes in key recruitment metrics and overall clinical trial development measures in industry-sponsored phase III clinical trials with a drug as an intervention conducted globally during 2008–2019.

## Methods

Our primary aim was to investigate whether there has been a significant change in key recruitment performance metrics in industry-sponsored phase III clinical trials with a drug as an intervention conducted globally when comparing three time intervals during 2008–2019. Recruitment performance was evaluated by three key recruitment metrics: recruitment rate (number of participants enrolled per month of recruitment), the number of sites registered per clinical trial, and the number of participants enrolled per site. Our second aim was to examine clinical trial development measures (trial duration, number of participants, number of clinical trials) over time from 2008–2019, and to examine the distribution of these trials by geographic location of trial sites, number of geographic regions registered and therapeutic area.

### Study design and data extraction

We analysed all industry-sponsored phase III clinical trials registered in the clinical trials database ClinicalTrials.gov as completed during 2008–2019. Data were extracted on the 25th of January 2021 from a cloud-based access through R-studio to the Aggregate Analysis of ClinicalTrials.gov database (AACT), which is a publicly available database that contains all information about every study registered in ClinicalTrials.gov [20]. We identified all completed interventional trials that were classified as phase III or phase II/III, and were funded by an industry-sponsor. In September 2007, the Food and Drug Administration (FDA) Amendments Act was signed into law. All clinical trials of FDA-regulated drugs or biological products that are started or ongoing after 26th December 2007 must be registered in ClinicalTrials.gov (except phase I trials) [21]. Therefore, we included all trials completed from 2008. To investigate changes in key recruitment metrics and clinical trial development measures over time, we compared trials completed during three time intervals of 4 years each; from 2008–2011, from 2012–2015, and from 2016–2019.

### Classification of geographic region and therapeutic area

Geographic region was classified according to the United Nations' classification that includes five main regions [22]. We included the subdivision of the Americas and Europe resulting in nine regions: Africa, Asia, Latin America and the Caribbean, Northern America, Oceania, Eastern Europe, Northern Europe, Southern Europe, and Western Europe. As illustrated by Fig 1, not all trials had registered data on the location of trial sites. This data was found within "Recruitment Details" in the AACT database, resulting in data on 1117 trials. For each of the 1117 trials, we mapped data on trial sites' location by region. Trials were excluded if one of the following conditions were met: 1) no trial sites were registered, 2) a geographic location of trial

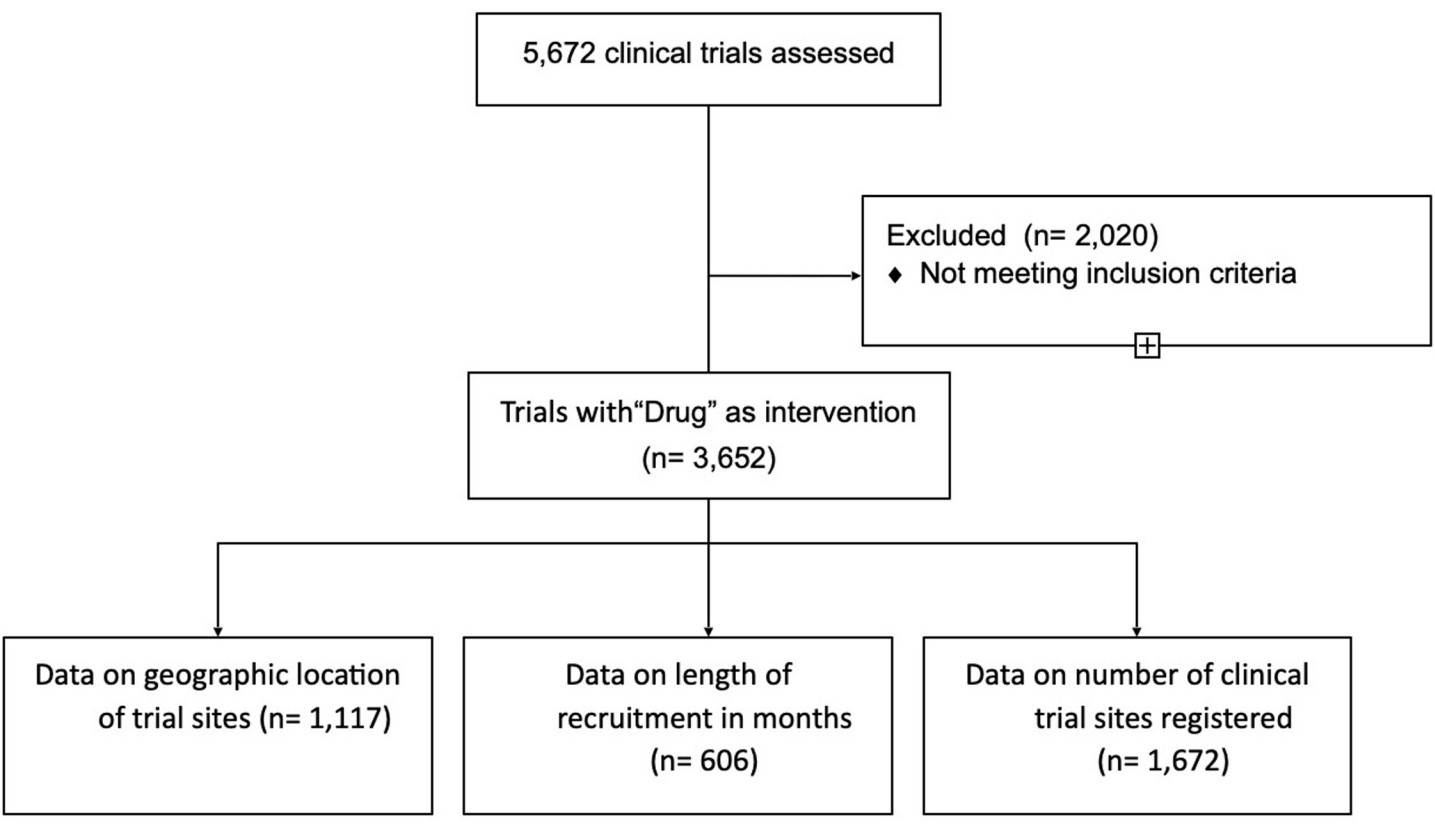

**Fig 1. CONSORT flow diagram.**

sites including overall continents such as "Asia, Europe and Australia" making it impossible to map the location within the 9 geographical regions.

Therapeutic area was classified according to the International Statistical Classification of Diseases and Related Health Problems (ICD-10) [23]. In total, 12 therapeutic areas were included: Certain infectious diseases, Circulatory system, Digestive system, Endocrine and metabolic system, Genitourinary system, Mental disorders, Musculoskeletal system, Neoplasms, Nervous system, Respiratory system, Skin, and Other therapeutic areas. In Clinical-Trials.gov, the therapeutic area of a clinical trial is listed in a medical condition description field. For each trial, we manually reviewed the medical condition description.

### Data analysis

All data analysis was conducted using R-language, and $p < 0.05$ was considered statistically significant. As the data on overall clinical trial development and key recruitment metrics collected from the AACT database were non-parametric data, Kruskal-Wallis tests were used to compare overall differences of means between the three groups. Further, Wilcoxon tests were used for pairwise tests for significance across the three groups. For geographical data, the distribution of trial sites and the mean number of geographic regions included in a trial were compared across the three time periods using the parametric Anova test as data was normally distributed with no extreme outliers. Further, the distribution of therapeutic areas across the three time periods was compared using parametric Anova tests.

## Results

As illustrated by the CONSORT flow diagram in **Fig 1**, 5672 clinical trials fulfilled the search criteria. Trials with invalid date registration, or invalid participant enrollment registration were excluded. For each of the 5672 trials we extracted data on the number of participants enrolled and trial duration in months. Out of the 5672 trials, 3652 trials had "drug" registered as intervention. For each of the 3652 trials we extracted data on the therapeutic area. Of these 3652 clinical trials, 1672 trials had valid data on the number of sites and 606 trials had valid data on the length of recruitment in months. Further, 1117 trials had data on geographic location of trial sites.

### Overall clinical trial development measures from 2008–2019

Of the 5,672 trials, 1,965 trials were completed from 2008–2011, 2,032 trials were completed from 2012–2015, and 1,676 trials were completed from 2016–2019. We found a decrease in the number of industry-sponsored phase III clinical trials conducted globally during 2008–2019 (see appendices, Table 1). From 2008–2011, 1965 trials were completed, and from 2016–2019, 1,676 trials were completed, corresponding to a decrease of 15% over time. The highest number of trials (2,031) was completed during 2012–2015. Similarly, we found an overall decrease in the number of participants enrolled in clinical trials during 2008–2019, again with the highest number of enrolled participants (1,249,809) registered between 2012–2015. From 2016–2019, 1,156,515 participants were enrolled, corresponding to a decrease of 7% compared to 2012–2015. However, as shown in Fig 2A, this decrease was only significant between the two time intervals from 2012–2015 and 2016–2019 ($p = 0.046$). Overall, we found no significant difference in the number of participants enrolled during 2008–2019 ($p = 0.1024$). For trial duration in months, a significant increase was found between all three time intervals from 2008–2019 ($p = <2.2e-16$). As seen in Fig 2B, trial duration increased significantly pairwise between all three time intervals. The overall median trial duration in months increased from 22 (InterQuartile Range (IQR) 13–35) in 2008–2011, to 24 (IQR 14–40, $p = 1.7e-05$) in 2012–2015, to a median trial duration of 28 months (IQR 17–47, $p = 9.3e-11$) in 2016–2019.

### Clinical trial key recruitment metrics from 2008–2019

From the 3,652 industry-sponsored phase III clinical trials with "drug" as intervention, 606 clinical trials had valid registration on length of recruitment. We found a significant increase in recruitment duration in months from an average recruitment period of 13 months (IQR 7–23) in 2008–2011 to 18 months (IQR 11–28) in 2016–2019 ($p = 0.0068$) (see appendices, Table 2). As shown in Fig 3A, no change in median recruitment duration in months was found between 2008–2011 and 2012–2015, however a significant increase in length of recruitment was found from 2012–2015 to 2016–2019 ($p = 0.0048$). For recruitment rate, we found a non-significant decrease in the number of participants enrolled per month of recruitment from an average of 26 participants (IQR 10–60) in 2008–2011 to 20 participants (IQR 10–46)

**Table 1. Basic characteristics of industry-sponsored phase III trials conducted during 2008–2019.** Over time development of number of trials, number of participants enrolled and trial duration in months are presented in the table together with corresponding p-values.

| Trial characteristics | Trials completed 2008–2011 | Trials completed 2012–2015 | Trials completed 2016–2019 | p-value (Kruskal-Wallis test**) |
|---|---|---|---|---|
| Number of trials | 1965 | 2031 | 1976 | |
| Total number of participants enrolled | 1.245.175 | 1.249.809 | 1.156.515 | P = 0.1024 |
| Trial duration in months, median (IQR) | 22 (13–35) | 24 (14–40) | 28 (17–47) | P<2.22e-16 |

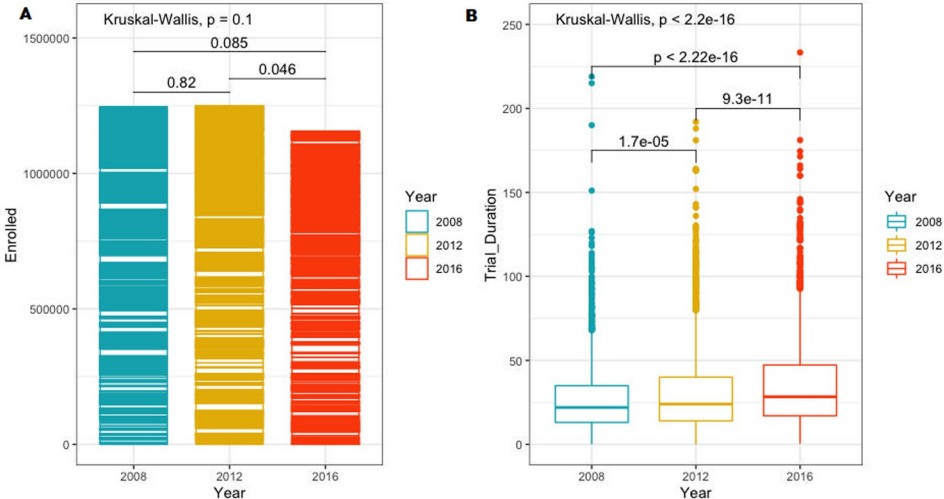

**Fig 2.** Comparison of clinical trial development metrics across the three time periods from 2008–2019 (A: Comparison of the mean number of participants enrolled in clinical trials from 2008–2019. B: Comparison of mean trial duration in months from 2009–2019).

in 2012–2016 (p = 0.195). As illustrated in Fig 3B, no significant difference in recruitment rate was found pairwise across the three groups. From the 3652 phase III clinical trials with "drug" as intervention, 1672 clinical trials had registered valid data on the number of sites per clinical trial in ClinicalTrials.gov. As shown in Fig 3C, we found a significant increase in the median number of sites registered per clinical trial from 43 sites (IQR 17–84) in 2012–2015 to 64 sites (IQR 30–118) in 2016–2019 (p = 0.025). In 2008–2011 the median number of sites per clinical trial was 51 (IQR 24–93), and consequently, the increase in the average number of sites across all three time intervals was non-significant (p = 0.077).

For the development of site effectiveness over time from 2008–2019, we investigated two metrics; 1) number of participants enrolled per site, and 2) the number of participants enrolled per site per month. As illustrated in Fig 3D, across all three time intervals from 2008–2019, we found a significant decrease in the overall median number of participants enrolled per site (p = 2.8e-06). In 2008–2011, sites enrolled 7 participants (IQR 5–11) on average, while this number decreased to an average of 6 participants (IQR 4–10) in 2016–2019. In 2012–2015, the median number of participants enrolled per site was also 7 (IQR 5–13). For the median number of participants enrolled per site per month, we found a non-significant decrease from 0.8 participants (IQR 0.3–1.9) in 2012–2015 to 0.4 (IQR 0.2–0.6) in 2016–2019. As shown in Fig

**Table 2. Key recruitment characteristics of industry-sponsored phase III trials with "drug" as intervention conducted during 2008–2019.** Over time development of recruitment duration, recruitment rate, number of sites and site effectiveness are presented in the table together with corresponding p-values.

| Trial characteristics | Trials completed 2008–2011 | Trials completed 2012–2015 | Trials completed 2016–2019 | p-value (Kruskal-Wallis test**) |
|---|---|---|---|---|
| Recruitment duration in months, median (IQR) | 13 (7–23) | 13 (8–22) | 18 (11–28) | P = 0.0068 |
| Number of participants enrolled per month of recruitment, median (IQR) | 26 (10–60) | 23 (8–61) | 20 (10–46) | P = 0.37 |
| Number of sites, median (IQR) | 51 (24–93) | 43 (17–84) | 64 (30–118) | P = 3.945e-08 |
| Number of participants enrolled per site, median (IQR) | 7 (5–11) | 7 (5–13) | 6 (4–10) | P = 2.755e-06 |
| Number of participants enrolled per site per month, median (IQR) | 0.6 | 0.8 | 0.4 | P = 5.841e-06 |

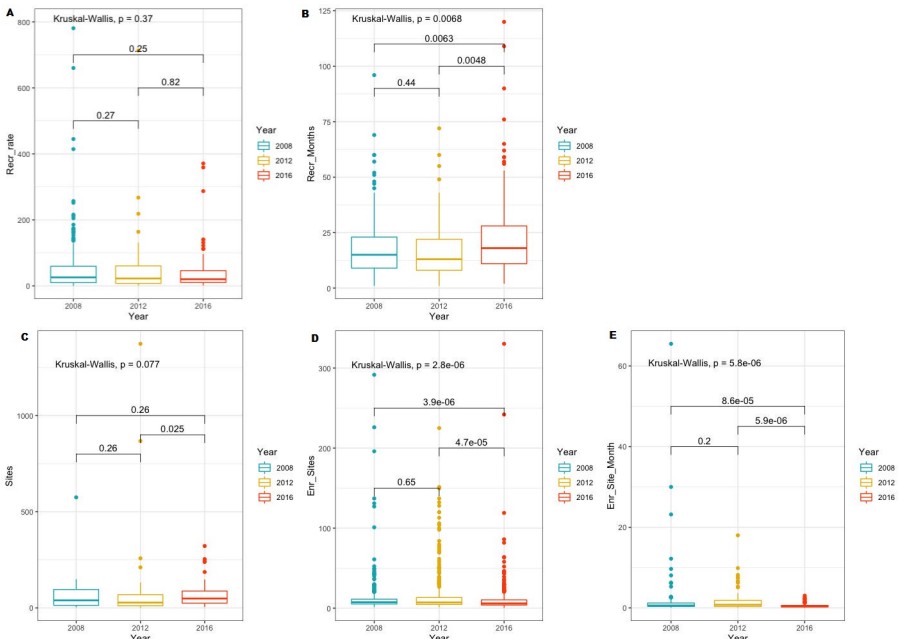

**Fig 3.** Comparison of key recruitment metrics across the three time periods from 2008–2019 (A: Comparison of recruitment rate (mean, IQR), B: Comparison of recruitment duration in months (mean, IQR), C: Comparison of the number of sites (mean, IQR), D: Comparison of the number of participants enrolled per site (mean, IQR), E: Comparison of the number of participants enrolled per site per month (mean, IQR)).

3E, no significant differences in the number of participants enrolled per site per month was found pairwise across all three groups.

## Geographic region and therapeutic area

For the development of clinical trial site location over time during 2008–2019, we found a significant decrease in the number of trial sites located in North America (p = 0.03), as illustrated in Fig 4A. Further, we found a significant increase in the number of trial sites located in Northern Europe (p = 0.006), Western Europe (p = 0.001), Southern Europe (p = 6.03e-06), Eastern Europe (p = 3.02e-08), Asia/Pacific (p = 4.2e-15), and Oceania (p = 0.0007). We found non-siginicant increases across the three time periods for the remaining geographic regions, including Latin America and Africa (see appendices, Table 3). For the mean number of geographic regions included in a clinical trial, we found an overall significant increase (p = 7.66e-07) when comparing the three time periods from 2008–2019. As shown in Fig 5, a significant increase in the mean number of regions was also found between the two time periods from 2012–2019 (p = 0.0018).

Fig 6 illustrates the distribution of therapeutic areas over time. We found overall significant decreases throughout the three time periods for trials examining digestive system (p = 0.0001), endocrine/metabolic system (p = 0.004), genitourinary system (p = 0.02), mental disorders (p = 2.2e-06), and nervous system (p = 0.01) (see appendices, Table 4). Overall significant increases were found for trials examining neoplasms (p = 1.03e-12) and skin (p = 8.3e-16). For infections, circulatory system and musculoskeletal system, we found no significant changes from 2008–2019.

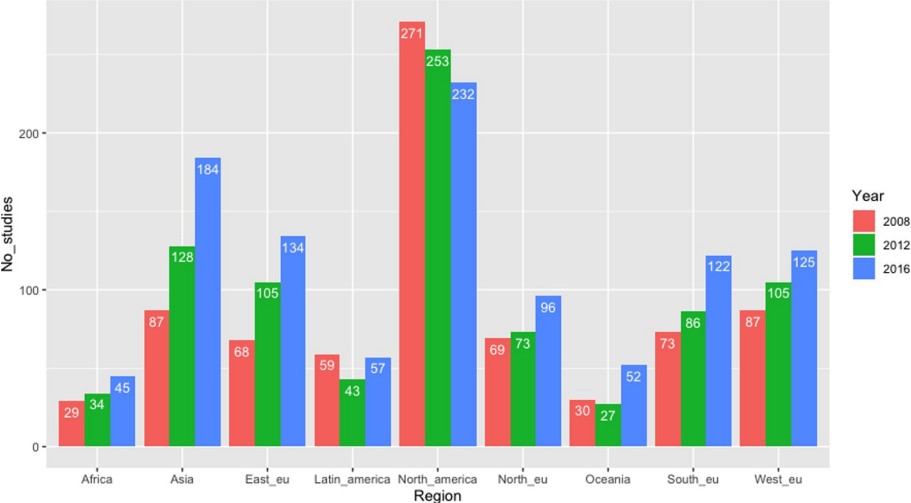

**Fig 4. Development of geographic location of clinical trials from 2008–2019.**

## Discussion

Our study shows a decline in several key recruitment performance metrics for industry-sponsored phase III clinical trials with a drug as an intervention conducted globally from 2008–2019. From the trials included in this study, we found that recruitment duration as well as overall clinical trial duration for phase III clinical trials have increased significantly during the last 12 years. Further, we found that industry-sponsored phase III clinical trials have increased the number of registered sites per clinical trial by more than 30% during the last 12 years, and concurrently, the number of participants enrolled in clinical research has decreased significantly. These findings indicate that recruitment for phase III clinical trials is less effective today compared to 12 years ago. The median number of participants enrolled per site in our study ranged from 7 (IQR 5–11) in 2008–2011 to 6 (IQR 4–10) in 2016–2019, reflecting the findings from The European Federation of Pharmaceutical Industries and Associations (EFPIA), who found that 8.8 participants were enrolled on average per active site in 931 industry-sponsored phase II-III clinical trials [24]. For the number of participants enrolled per month of recruitment (recruitment rate) and the number of participants enrolled per site (or per site per month), we found non-significant decreases.

**Table 3. Development of geographic location of clinical trial sites from 2008–2019 spread over 9 geographic regions including North America, Latin America, Northern Europe, Western Europe, Southern Europe, Asia/Pacific, Africa and Oceania.** Each clinical trial may have registered several geographical regions.

| Geographic region | Trials completed 2008–2011 | Trials completed 2012–2015 | Trials completed 2016–2019 | p-value |
|---|---|---|---|---|
| North America (%) | 271 (73) | 253 (65) | 232 (65) | 0.03287 |
| Latin America (%) | 59 (16) | 43 (11) | 57 (16) | 0.07967 |
| Northern Europe (%) | 69 (18.5) | 73 (19) | 96 (27) | 0.006061 |
| Western Europe (%) | 87 (24) | 105 (27) | 125 (35) | 0.001416 |
| Southern Europe (%) | 73 (20) | 86 (22) | 122 (34) | 6.03e-06 |
| Eastern Europe (%) | 68 (18) | 105 (27) | 134 (38) | 3.02e-08 |
| Asia/Pacific (%) | 87 (23) | 128 (33) | 184 (52) | 4.21e-15 |
| Africa (%) | 29 (8) | 34 (9) | 45 (13) | 0.06161 |
| Oceania (%) | 30 (8) | 27 (7) | 52 (15) | 0.0007428 |

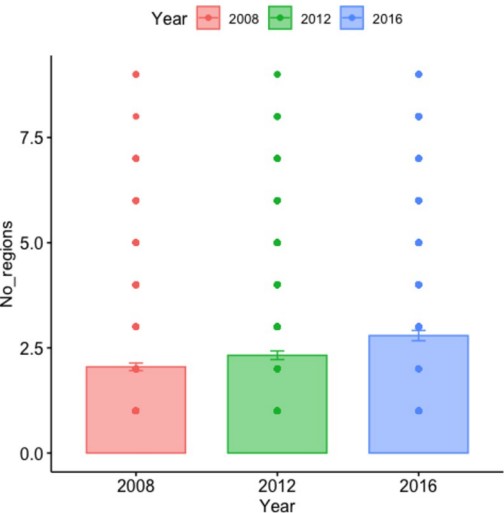

**Fig 5. Development of the number of regions registered per clinical trial over time from 2008–2019.**

The combination of increasing number of sites in clinical trials and decreasing participant enrollment could be due to more sites being low- or non-performing, as more sites are needed to comply with timelines and sample sizes. As previously mentioned, clinical trials are facing challenges recruiting more narrowly defined patient subpopulations, as new technologies and diagnostic tools have paved the way for diagnosing and treating chronic, difficult-to-treat and rare diseases [3, 25, 26]. This might be one explanation to the seemingly decrease of recruitment and site effectiveness found in this study, especially during the last 8 years. However, an increase in the number of sites per clinical trial could also be an expression of an increased globalisation. In this study, we found an overall significant increase in the mean number of regions included in clinical trials from 2008–2019. Further, our findings indicate a shift in location of clinical trial sites towards emerging regions such as Eastern Europe and Asia/Pacific with a concurrent decrease in the number of trial sites located in North America from 2008–2019. This reflects the findings from several studies investigating the increasing

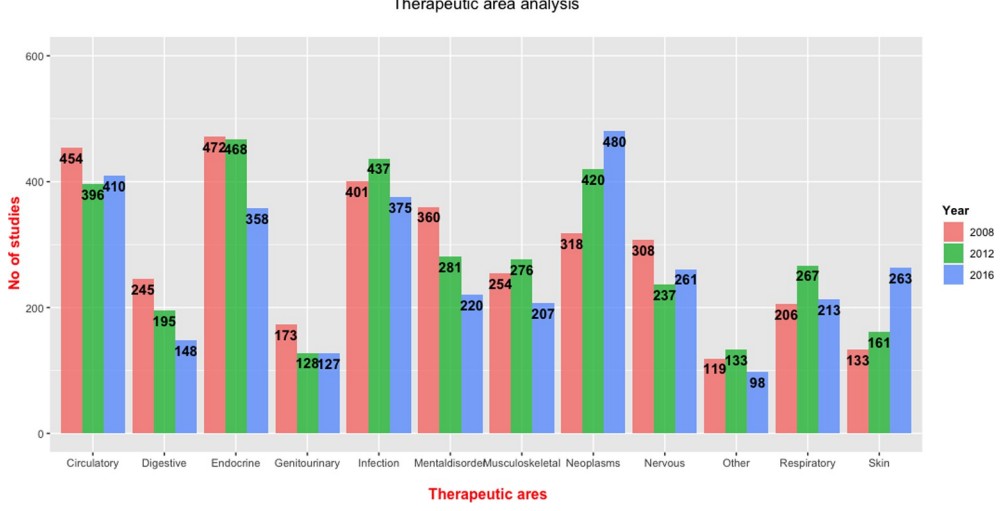

**Fig 6. Distribution of therapeutic area in clinical trials from 2008–2019.**

**Table 4. Development of therapeutic area investigated in clinical trials completed from 2008–2019.**

| Therapeutic area | Trials completed 2008–2011 | Trials completed 2012–2015 | Trials completed 2016–2019 | p-value |
|---|---|---|---|---|
| Infections (%) | 401 (12) | 437 (13) | 375 (12) | 0.2669 |
| Circulatory system (%) | 454 (13) | 396 (12) | 410 (13) | 0.1183 |
| Digestive system (%) | 245 (7) | 195 (6) | 148 (5) | 0.0001471 |
| Endocrine/metabolic system (%) | 472 (14) | 468 (14) | 358 (11) | 0.004219 |
| Genitourinary system (%) | 173 (5) | 128 (4) | 127 (4) | 0.02568 |
| Mental disorders (%) | 360 (10.5) | 381 (11) | 220 (7) | 2.15e-06 |
| Musculoskeletal system (%) | 254 (7) | 276 (8) | 207 (6.5) | 0.05394 |
| Neoplasms (%) | 318 (9) | 420 (12) | 480 (15) | 1.03e-12 |
| Nervous system (%) | 308 (9) | 237 (7) | 261 (8) | 0.009974 |
| Respiratory system (%) | 206 (6) | 267 (8) | 213 (7) | 0.008613 |
| Skin (%) | 133 (4) | 161 (5) | 263 (8) | 8.28e-16 |
| Other (%) | 119 (3) | 133 (4) | 98 (3) | 0.2026 |

globalisation of clinical trials [27–29], reporting that the number of clinical trials conducted outside of the United States and Europe have doubled over the past 10 years [30]. There are many factors influencing the selection of clinical trial sites, such as operational costs and the regulatory landscape in different markets. However, the opportunity for increased exposure towards large, untested research populations could also be an important factor contributing to the increased globalisation [27].

## Recruitment

For clinical trial recruitment performance, increasing complexity in clinical trials [26] and patients' willingness to participate in clinical research [31, 32] are two important factors that seem to reinforce each other.

Designing protocols that balance measuring efficacy and safety with minimized patient participation burden and advanced levels of individually targeted therapies, is a difficult task [33]. In 2017, Getz. et al found an increasing trend in clinical trial protocol complexity across phase I, II and III clinical trials based on 9373 protocols of which 76% were protocols provided by large companies [3]. For phase III trials, they found a relative growth of 70% in total procedures carried out as well as an increase of 25% in planned study visits. Further, Getz KA et al. found a growth of 34% in mean cost per visit over the last 10 years. The mean total cost per study volunteer per study increased substantially even though the cost for many study procedures, such as blood tests, has decreased during the past decade. These findings could pose a threat to the effectiveness of patient recruitment and retention and could lead to increasing numbers of protocol amendments, adding to recruitment and thereby timeline delays. As the number of procedures increases, the work required to support these increases similarly [34]. Furthermore, demanding protocols could also negatively influence clinical trial site performance, increasing the level of stress for trial site personnel [35]. High complexity and demanding trial site tasks may impact the performance of clinical trial sites, where low- and non-performing sites are already a challenge for the clinical trial industry [11]. In an analysis from 2018 conducted by the Boston Consulting Group and KMR Group, data from more than 75000 sites were used to examine site performance and underlying drivers of success. The analysis concluded that companies that reduce the number of non-enrolling sites by approximately 50% save around $10–13 million on an average phase III trial [13].

For trial participants, the willingness to enter and complete clinical trials is negatively correlated to a high number of study visits and performed procedures as well as more

comprehensive and advanced consent forms [32, 36]. Patients participate in clinical trials for a number of reasons, ranging from receiving new medication not yet available for the general population, to making a contribution that could help another person [37]. Patient-centricity is an accelerating trend in clinical trials, and transparent patient reimbursement is an important part of this. The increasing trial complexity with clinical trials often being carried out at specialist centers, requiring patients to travel longer and perhaps also more often, has put a new focus on patient reimbursement [38]. Compensating patients in clinical trials may be seen as an award for a risk taken by the patient, however reimbursement rates range from approximately 400$ to 2,000$ per patient in clinical trials, significantly impacting the costs of clinical research [39]. Increasing patient reimbursement rates may be necessary due to the increase in trial complexity, however it may also pose a threat to patient willingness to participate in clinical trials, nudging patients towards trials with higher levels of compensation.

Increasing cost of clinical research may have significant implications for public health, as it affects drug companies' willingness to undertake clinical trials, which in turn limits patient access to novel treatments [17]. Armed with such knowledge, a growing number of pharmaceutical companies and contract research organizations (CRO's) are working to optimize recruitment for clinical trials through optimizing study designs in order to improve feasibility and ease site and subject participation burden [40]. The declining productivity of drug development has intensified the interest in digital tools and other technologies that may improve trial efficiency [11]. In a systematic review and meta analysis from 2019, we found that targeting participants using online remedies was an effective tool for patient recruitment in clinical trials, as online recruitment was superior to traditional in-clinic recrutiment in regards to both time-efficiency and cost-effectiveness [41]. However, our study also concluded that for online recruitment of participants to be an effective tool, both effort and money should be invested in recruitment ads and campaigns as well as in training trial staff. With fewer trial sites, logistical complexity could be limited and effective recruitment, either through high-performing sites or through digital tools, are a key factor in minimizing recruitment delays. This will in turn impact cost-per-patient and time-to-market, increasing the sponsor's return on investment, increasing value from the drugs patent lifespan and the societal benefit of bringing new therapies to patients in need [11].

## Risk of bias

The results reported in this study rely on voluntary reporting from trial sponsors, and therefore could be biased. Further, not all clinical trials are registered in ClinicalTrials.gov and the registration compliance probably varies across countries. However, it is mandatory to register all phase III clinical trials of FDA-related drugs and biological products in ClinicalTrials.gov [21] and hence, we expect that the majority of industry-sponsored phase III clinical trials are registered. Moreover, since 2005, the International Committee of Medical Journal Editors (ICMJE) has required prospective registration of clinical trials in ClinicalTrials.gov or the WHO International Clinical Trials Registry Platform (ICTRP) as a precondition for publication [42]. The norms and rules for reporting trials in ClinicalTrials.gov change over time. The overall clinical trial metrics included in this study, such as the number of participants enrolled, is mandatory upon trial registration and reporting of results. However, for recruitment metrics such as recruitment duration and geographic location of trial sites, manual reporting is required for trial sponsors. Accordingly, the risk of bias mostly relates to non-mandatory metrics where manual reporting without clear guidelines is required. Further, the geographic distribution of trials could be biased due to unequal registration of trials in different geographic regions.

## Study limitations

This study on changes in recruitment performance over time has several limitations. Notably, key recruitment metrics including recruitment duration in months and the number of sites per clinical trial are registered manually as a free text by trial sponsors under "recruitment details" in the AACT database. Hereby, far from every trial had data on recruitment and site metrics, reducing the scope of the dataset in this study. There is a risk that the sponsors are more likely to fill in the free text under "recruitment details" making the results less representative from a global perspective. For the assessment of site effectiveness, it was impossible to estimate the number of sites that did not enroll any participants as these data are not registered in ClinicalTrials.gov. Hence, no associations between geographic location of sites and site performance could be investigated in this study. Further, we only collected data from clinical trials that were completed within the three defined time periods and consequently, no data on recruitment from all ongoing and not yet completed clinical trials is included in this study.

Lastly, neoplasms are the largest therapeutic area in the last period from 2016–2019. In the field of cancer there has been a change in phase 3 trials due to targeted treatment and companion diagnostics. Phase 3 trials are more biomarker-driven which means that large patient groups no longer exist, and large phase 3 trials are impossible to complete. Even frequently occurring cancers are divided into many rare subtypes. This may possibly affect the outcome. Further, the results show a trend with an overall reduction in multicenter trials. However, we have not investigated the trends for all industry sponsored trials.

## Conclusion

We assessed changes in recruitment performance over time in industry-sponsored phase III clinical trials and found a decline in key recruitment metrics. Recruitment duration have increased significantly during the last 12 years from an average recruitment period of 13 months in 2008–2011 to 18 months in 2016–2019. Further, the number of registered sites per clinical trial has increased by more than 30% during the last 12 years, and the number of participants enrolled has decreased significantly.

Clinical trials have become larger measured on the number of sites registered per clinical trials, and recruitment for clinical trials takes more time and less participants are enrolled, indicating that recruitment for phase III clinical trials is less effective today compared to 12 years ago.

For future research, more and deeper knowledge of which factors are strongly associated with unsuccessful recruitment is essential to identify suitable strategies to improve recruitment.

## Author Contributions

**Conceptualization:** Simon Francis Thomsen.

**Data curation:** Mette Brøgger-Mikkelsen.

**Formal analysis:** Mette Brøgger-Mikkelsen.

**Investigation:** Mette Brøgger-Mikkelsen.

**Supervision:** Zarqa Ali, Simon Francis Thomsen.

**Visualization:** Mette Brøgger-Mikkelsen.

**Writing – original draft:** Mette Brøgger-Mikkelsen.

**Writing – review & editing:** John Robert Zibert, Anders Daniel Andersen, Ulrik Lassen, Merete Hædersdal, Zarqa Ali, Simon Francis Thomsen.

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
