## [Decision Letter · Decision Letter 0]

16 May 2022

PONE-D-22-12003Changes in Key Recruitment Performance Metrics from 2008-2019 in Industry-sponsored Phase III Clinical Trials Registered at ClinicalTrials.govPLOS ONE

Dear Dr. Ali,

Thank you for submitting your manuscript to PLOS ONE. After careful consideration, we feel that it has merit but does not fully meet PLOS ONE’s publication criteria as it currently stands. Therefore, we invite you to submit a revised version of the manuscript that addresses the points raised during the review process.

We look forward to receiving your revised manuscript.

Kind regards,

Sathishkumar V E

Academic Editor

PLOS ONE

Journal Requirements:

"No funders had no role in study design, data collection and analysis, decision to publish, or preparation of the manuscript."

4. Please upload a new copy of Figures 3, 4 and 6 as the detail is not clear. Please follow the link for more information: https://blogs.plos.org/plos/2019/06/looking-good-tips-for-creating-your-plos-figures-graphics/"" https://blogs.plos.org/plos/2019/06/looking-good-tips-for-creating-your-plos-figures-graphics/

Reviewers' comments:

Reviewer's Responses to Questions

**Comments to the Author**

1. Is the manuscript technically sound, and do the data support the conclusions?

Reviewer #1: Yes

Reviewer #2: Yes

2. Has the statistical analysis been performed appropriately and rigorously? 

Reviewer #1: Yes

Reviewer #2: Yes

3. Have the authors made all data underlying the findings in their manuscript fully available?

Reviewer #1: Yes

Reviewer #2: Yes

4. Is the manuscript presented in an intelligible fashion and written in standard English?

Reviewer #1: Yes

Reviewer #2: Yes

5. Review Comments to the Author

Reviewer #1: Comments to authors

Interesting paper but I could do with less statistical analysis and significance testing.

The generic question of interest is whether the drug industry is doing less trials over time. That is true from your results for multicenter trials but is it true overall for all industry sponsored trials?

My horseback counts from ClinicalTrials.gov for industry sponsored trials completed 2008-2019 is 64,537 and 15,517 for multicenter trials. Counts corresponding to your Table 1 for all industry sponsored trials are 20,092, 21,297, and 23,146 and for multicenter trials were 4,633, 5,090, and 5,794. You would do well with an introduction as to how you get from these numbers to those focused upon in your paper and also if these results reinforce your conclusions.

Curtis Meinert

Reviewer #2: Add related works as a separate section

Contributions of the research can be added at the end of introduction section

Conclusion should be added at the end of the manuscript providing overall summary of the manuscript

Provide clear images for all the Figures

Line 188, As seen in figure 2b, trial duration increased significantly pairwise between all three time intervals. Significantly pairwise?

If possible add Exploratory Data Analysis

Which users benefit from this study?

6. PLOS authors have the option to publish the peer review history of their article (what does this mean?). If published, this will include your full peer review and any attached files.

Reviewer #1: No

Reviewer #2: **Yes: **Usha Moorthy

---

## [Author Response · Author response to Decision Letter 0]

28 Jun 2022

Re: PONE-D-22-12003 “Changes in Key Recruitment Performance Metrics from 2008-2019 in Industry-sponsored Phase III Clinical Trials Registered at ClinicalTrials.gov” by Brøgger-Mikkelsen et al.

Dear Editor-in-Chief 

Thank you for giving us the opportunity to revise our paper and for the valuable suggestions from the reviewers. We have revised the paper according to the suggestions and issues raised.

Comment 1:

Author response: 

The manuscript has been revised carefully to meet PLOS ONE’s style requirements. We hope the manuscript is suitable for PLOS ONE now.

Comment 2:

Thank you for stating the following financial disclosure: 

"No funders had any role in study design, data collection and analysis, decision to publish, or preparation of the manuscript."

Author response: 

The authors received no specific funding for this work. This statement has been added to the cover letter and thank you for changing the online submission on our behalf. 

Comment 3:

We note that you have indicated that data from this study are available upon request. PLOS only allows data to be available upon request if there are legal or ethical restrictions on sharing data publicly. For more information on unacceptable data access restrictions, please see http://journals.plos.org/plosone/s/data-availability#loc-unacceptable-data-access-restrictions. 

Author response: 

The link to all raw data is https://aact.ctti-clinicaltrials.org/connect. All the data is publicly available and with use of the provided link it is possible to get access to data used for analyses in the present manuscript. 

Comment 4:

Please upload a new copy of Figures 3, 4 and 6 as the detail is not clear. Please follow the link for more information: https://blogs.plos.org/plos/2019/06/looking-good-tips-for-creating-your-plos-figures-graphics/"" https://blogs.plos.org/plos/2019/06/looking-good-tips-for-creating-your-plos-figures-graphics/

Author response: 

A new copy of figure 3, 4, and 6 has been provided in the required format. 

Comment 5:

Author response: 

The reference list has been reviewed to ensure it is complete and correct.

Reviewers' comments:

Reviewer #1: 

Comment 1:

Interesting paper but I could do with less statistical analysis and significance testing.

The generic question of interest is whether the drug industry is doing less trials over time. That is true from your results for multicenter trials but is it true overall for all industry sponsored trials?

Author response: 

Thanks for rising this interesting comment. Our results suggest that there is a reduction in multicenter trials; however, this is not true for all industry sponsored trials. It would be interesting to examine that more in the future. We have elaborated this issue in the section of limitation.

Comment 2:

My horseback counts from ClinicalTrials.gov for industry sponsored trials completed 2008-2019 is 64,537 and 15,517 for multicenter trials. Counts corresponding to your Table 1 for all industry sponsored trials are 20,092, 21,297, and 23,146 and for multicenter trials were 4,633, 5,090, and 5,794. You would do well with an introduction as to how you get from these numbers to those focused upon in your paper and also if these results reinforce your conclusions.

Author response: 

Thanks for this comment. We have not included all industry-sponsored trials but only phase III trials with a drug as an intervention. We have emphasized that in the last section of introduction, first section of methods, and first section of discussion. 

Reviewer #2: 

Comment 1:

Add related works as a separate section

Author response: 

Related work has been added as a separate section in the discussion.

Comment 2:

Contributions of the research can be added at the end of introduction section

Author response: 

Author contribution has been added at the end of introduction. 

Comment 3:

Conclusion should be added at the end of the manuscript providing overall summary of the manuscript

Author response: 

Thank you for this comment. Conclusion section has been added at the end of the manuscript and revised to provide an overall summary of the manuscript. 

Comment 4:

Provide clear images for all the Figures

Author response: 

Thanks for making us ware of it. Clear images for all the figures have been provided. 

Comment 5:

Line 188, As seen in figure 2b, trial duration increased significantly pairwise between all three time intervals. Significantly pairwise?

If possible add Exploratory Data Analysis

Author response: 

Thank you for this comment. We acknowledge that the sentence may be confusing. The sentence has been rephrased and “pairwise” has been deleted.

Further, we recognize the merit of exploratory data analysis, but for the purpose of this study, we believe that our aggregate data analysis in the tables and figures represents data well.

Comment 6:

Which users benefit from this study?

Author response: 

Thanks for this interesting question. Researchers and academics trying to understand recruitment patterns would benefit from this work. Private recruitment companies will also benefit from better understanding of changes in the recruitment over time. 

Looking forward to hearing from you,

Yours sincerely

Zarqa Ali, MD PhD

Dept. of Dermatology

Bispebjerg Hospital

DK-2400 København NV

Zarqa_ali@hotmail.com

---

## [Decision Letter · Decision Letter 1]

8 Jul 2022

Changes in Key Recruitment Performance Metrics from 2008-2019 in Industry-sponsored Phase III Clinical Trials Registered at ClinicalTrials.gov

PONE-D-22-12003R1

Dear Dr. Ali,

We’re pleased to inform you that your manuscript has been judged scientifically suitable for publication and will be formally accepted for publication once it meets all outstanding technical requirements.

Kind regards,

Sathishkumar V E

Academic Editor

PLOS ONE

Additional Editor Comments (optional):

Reviewers' comments:

Reviewer's Responses to Questions

**Comments to the Author**

1. If the authors have adequately addressed your comments raised in a previous round of review and you feel that this manuscript is now acceptable for publication, you may indicate that here to bypass the “Comments to the Author” section, enter your conflict of interest statement in the “Confidential to Editor” section, and submit your "Accept" recommendation.

Reviewer #2: (No Response)

2. Is the manuscript technically sound, and do the data support the conclusions?

Reviewer #2: (No Response)

3. Has the statistical analysis been performed appropriately and rigorously? 

Reviewer #2: (No Response)

4. Have the authors made all data underlying the findings in their manuscript fully available?

Reviewer #2: (No Response)

5. Is the manuscript presented in an intelligible fashion and written in standard English?

Reviewer #2: (No Response)

6. Review Comments to the Author

Reviewer #2: (No Response)

7. PLOS authors have the option to publish the peer review history of their article (what does this mean?). If published, this will include your full peer review and any attached files.

Reviewer #2: **Yes: **Usha Moorthy

---

## [Editor Report · Acceptance letter]

15 Jul 2022

PONE-D-22-12003R1 

Changes in Key Recruitment Performance Metrics from 2008-2019 in Industry-sponsored Phase III Clinical Trials Registered at ClinicalTrials.gov 

Dear Dr. Ali:

I'm pleased to inform you that your manuscript has been deemed suitable for publication in PLOS ONE. Congratulations! Your manuscript is now with our production department. 

Kind regards, 

on behalf of

Dr. Sathishkumar V E 

Academic Editor

PLOS ONE